# Interpretable machine learning for predicting delays in seeking abortion among reproductive-aged women in Ethiopia: A study using EDHS 2016 data

Meron Asmamaw Alemayehu[1]*, Almaw Genet Yeshiwas[2], Abebaw Molla Kebede[3], Habitamu Mekonen[4], Getaneh Atikilt Yemata[5], Amare Genetu Ejigu[6], Ahmed Fentaw Ahmed[2], Abathun Temesgen[2], Gashaw Melkie Bayeh[2], Chalachew Yenew[7], Rahel Mulatie Anteneh[5], Zeamanuel Anteneh Yigzaw[8], Fantu Mamo Aragaw[1], Getasew Yirdaw[9], Sintayehu Simie Tsega[10], Anley Shiferaw Enawgaw[11], Tilahun Degu Tsega[3,12], Zufan Alamrie Asmare[13], Berhanu Abebaw Mekonnen[14]

1 Department of Epidemiology and Biostatistics, Institute of Public Health, College of Medicine and Health Sciences, University of Gondar, Gondar, Ethiopia, 2 Department of environmental health, college of medicine and health science, Injibara university, Injibara, Ethiopia, 3 Department of Public Health, College of Medicine and Health Sciences, Injibara University, Injibara, Ethiopia, 4 Department of Human Nutrition, College of Health Science, Debre Markos University, Debre Markos, Ethiopia, 5 Depatment of Public Health, College of health science, Debre Tabor University, Debre Tabor, Ethiopia, 6 Department of Midwifery, College of Medicine and Health Sciences, Injibara University, Injibara, Ethiopia, 7 Department of Environmental Health Sciences, Public Health, College of Health Sciences, Debre Tabor University, Debre Tabor, Ethiopia, AND Research Degree PhD student at Jockey Club College of Veterinary Medicine and Life Sciences, City University of Hong Kong, Hong Kong SAR, China, 8 Department of Health Promotion and Behavioral Sciences, School of Public Health, College of Medicine and Health Sciences, Bahir Dar University, Bahir Dar, Ethiopia, 9 Department of Environmental Health Science, College of Medicine and Health Sciences, Debre Markos University, Debre Markos, Ethiopia, 10 Department of Medical Nursing, School of Nursing, College of Medicine and Health Science, University of Gondar, Gondar, Ethiopia, 11 Department of Public Health, College of Health Sciences, Debre Markos University, Debre Markos, Ethiopia, 12 AND Department of Epidemiology, School of Public Health, Cheeloo College of Medicine, Shandong University, Jinan, China 13 Department of Ophthalmology, School of Medicine and Health Science, Debre Tabor University, Debre Tabor, Ethiopia, 14 Department of Nutrition and Dietetics, School of Public Health, College of Medicine and Health Sciences, Bahir Dar University, Bahir Dar, Ethiopia

* merryalem101@gmail.com

## Abstract

Delayed access to abortion care in Ethiopia poses significant public health risks, yet it has not been studied using advanced machine learning models with interpretable techniques. This study aims to identify its key predictors through Shapley Additive Explanations (SHAP) values. The study used data from the 2016 Ethiopian Demographic and Health Survey. Data preprocessing tasks such as feature engineering, lumping, filtering, and encoding were performed before model building. Eight machine learning models, including LightGBM, Support Vector Machine, Random Forest, and XGBoost, were employed to predict delays in seeking an abortion. SHAP analysis was used to interpret feature importance and understand individual variable contributions. The prevalence of delayed abortion seeking was 1109 (54.3%). The

**Data availability statement:** The dataset used for this study is publicly available at the MEASURE DHS program website (https://www.dhsprogram.com/data).

**Funding:** The author(s) received no specific funding for this work.

**Competing interests:** The authors have declared that no competing interests exist.

**Abbreviations:** AUC, Area Under the Curve; CI, Confidence Interval; EDHS, Ethiopian Demographic and Health Survey; F1-Score, F1-Score; GBM, Gradient Boosting Machine; KNN, K-Nearest Neighbors; LR, Logistic Regression; NB, Naive Bayes; NPV, Negative Predictive Value; PPV, Positive Predictive Value; RF, Random Forest; ROC, Receiver Operating Characteristic; SHAP, Shapley Additive Explanations; SS, Sensitivity; SVM, Support Vector Machine; XGB, Extreme Gradient Boosting.

Random Forest model performed the best, with an accuracy of 91.8% (95% CI: 89.3, 93.8) and an AUC of 97.6, effectively predicting delays in abortion-seeking behavior. SHAP analysis revealed that age (women aged 40–49), regional factors (residing in the Somali and Amhara regions), and lack of media exposure were strong positive contributors to delays. In contrast, urban residence and living in Addis Ababa were associated with a lower likelihood of delay. Alcohol consumption also showed a positive association with delay. The study identifies key factors influencing delays in seeking abortion services in Ethiopia, highlighting the importance of targeted interventions, especially for older women and those in rural regions. These findings offer valuable insights for designing public health initiatives aimed at reducing unsafe abortion-related maternal morbidity and mortality.

## Author summary

Delays in seeking abortion care remain a major public health concern in Ethiopia, increasing the risk of complications and maternal death. Understanding why women delay seeking care is essential for designing effective interventions. In this study, we analyzed nationally representative data from Ethiopian women to identify the key factors associated with delayed abortion care. We applied advanced machine learning methods and used an interpretable approach to clearly explain how different factors influence delay. More than half of the women in our study experienced delays in seeking abortion services. We found that older age, living in certain regions such as Somali and Amhara, lack of media exposure, rural residence, and alcohol use were important factors associated with higher likelihood of delay. In contrast, living in urban areas, particularly in Addis Ababa, was linked to earlier care-seeking. By identifying the most influential predictors, our findings provide evidence that can help policymakers and public health professionals design targeted strategies to reduce delays and prevent complications related to unsafe abortion.

## Introduction

Access to safe and timely abortion services is a public health concern, particularly in low- and middle-income countries. Unsafe abortions contribute significantly to maternal morbidity and mortality, with an estimated 45% of all abortions worldwide being unsafe. In sub-Saharan Africa, and especially in Ethiopia, unsafe abortions account for a considerable portion of maternal deaths, despite efforts to improve access to reproductive healthcare. One of the primary factors contributing to unsafe abortion outcomes is the delay in seeking abortion services where abortion, as a time-sensitive healthcare service, should ideally be performed up to and including 12

weeks of gestation, before fetal viability. Delays beyond this period increase the risk of complications, particularly when procedures are performed later in pregnancy or under unsafe conditions [1,2].

In Ethiopia, various barriers continue to prevent women from seeking timely abortion care. These barriers include social stigma, economic challenges, limited access to healthcare facilities, particularly in rural areas, and restrictive legal frameworks (such as narrow legal grounds for eligibility, procedural requirements, and limited awareness of the legal provisions, all of which may discourage or delay women from seeking services in a timely manner). Understanding the factors associated with delayed abortion seeking is critical for developing targeted interventions to reduce unsafe abortions and improve maternal health outcomes. While previous research has identified socio-demographic factors such as educational attainment, marital status, residence, and economic hardship as contributors to these delays, the complexity of these interactions calls for more advanced analytical approaches [3,4].

Machine learning techniques offer powerful tools for analyzing large and complex datasets, providing the ability to capture non-linear relationships and interactions that traditional statistical methods may miss. However, many machine learning models are often criticized for their lack of interpretability, which limits their application in healthcare, where transparent decision-making is crucial. To address this limitation, this study employs interpretable machine learning techniques to predict delays in seeking abortion services among reproductive-age women in Ethiopia, using data from the 2016 Ethiopian Demographic and Health Survey (EDHS 2016). Interpretable models allow for not only more accurate predictions but also a clearer understanding of the relationships between different factors, making the results more actionable for healthcare providers and policymakers [4,5].

The primary objective of this study is to develop a machine learning model that accurately predicts the likelihood of delayed abortion seeking based on socio-economic, demographic, and health-related factors extracted from EDHS 2016 data. By focusing on interpretable machine learning approaches, such as Shapley Additive Explanations (SHAP), this study seeks to provide insights into the key predictors of delay and their interactions. This approach will help to identify the most significant barriers to timely abortion care in Ethiopia, offering valuable evidence for policymakers to design targeted interventions.

This study represents a novel application of interpretable machine learning to a major reproductive health problem in Ethiopia. We curated and analyzed the nationally representative EDHS 2016 dataset with a focus on abortion care, a domain where advanced analytics have been rarely applied. Methodologically, the study combined rigorous feature selection using the Boruta algorithm with model optimization and explainable AI techniques (SHAP) to balance predictive accuracy with transparency. By integrating these approaches, the study not only improves performance but also provides actionable insights into the determinants of delayed abortion care. The findings are expected to have significant implications for maternal health policies, enabling more effective strategies to reduce delays and promote safer abortion services across Ethiopia.

## Methods and materials

### Data source

This study used data from the 2016 Ethiopian Demographic and Health Survey (2016 EDHS) instead of the 2019 mini Ethiopian Demographic and Health Survey (2019 mEDHS). The main reason for this decision is that the 2019 mEDHS did not collect data on the target variable for this study, delayed abortion seeking. Since the 2016 EDHS includes relevant data on this topic, it was selected as the preferred source for analysis.

The 2016 EDHS was conducted between January 18 and June 27, 2016, across Ethiopia's nine regions and two city administrations. The survey aimed to generate nationally representative health data, with separate estimates for urban and rural areas as well as individual regions. A two-stage, residence-stratified sampling method was applied, resulting in the selection of 645 enumeration areas (EAs), 443 from rural and 202 from urban regions. Within each selected EA, households were enumerated, and in larger EAs with more than 300 households, one segment was randomly selected.

PLOS Digital Health

A fixed number of 28 households per cluster was then chosen using systematic random sampling, ensuring proportional representation across administrative levels.

The survey's questionnaires were administered to women and men aged 15–49 years who were either permanent residents or had stayed at least one night in the selected households. More details on the survey's methodology can be found in the official report [6].

## Modeling software

This study used R programming language (Version 4.2.1) for data preprocessing and machine learning implementation. Descriptive analysis, including frequencies and percentages of features, was also conducted with the same software. R packages were downloaded from the Comprehensive R Archive Network (CRAN), a global network that hosts the latest R code and documentation.

## Dataset and study variables

The study dataset included 2,042 women of reproductive age who sought abortion care. A range of sociodemographic, reproductive, household, and behavioral characteristics were captured to assess factors associated with delayed abortion seeking.

Sociodemographic variables included age (categorized into standard five-year reproductive age groups), geographic region (representing all administrative regions and city administrations), and place of residence (urban or rural). Educational attainment was grouped into four levels (no education, primary, secondary, and higher). Marital status categories included never in union, married, living with a partner, widowed, divorced, and separated. Women's occupational status was also recorded across common employment categories such as professional/technical/managerial, sales, agricultural employment, manual labor, and not working.

Household and partner-related characteristics included the husband's or partner's educational level, occupational category, and contraceptive decision-making dynamics (respondent-led, partner-led, joint decision-making, or other). The household wealth index, based on standard DHS composite measures, was categorized as poor, middle, or rich. Household composition variables included number of household members and number of under-five children.

Reproductive and behavioral characteristics included total number of children ever born, alcohol use, tobacco chewing/smoking, and exposure to mass media. These variables were categorized according to DHS standard definitions (e.g., alcohol use classified as yes/no; media exposure which is a composite measure classified as exposed or not exposed).

Together, these variables provided comprehensive coverage of women's sociodemographic backgrounds, their reproductive histories, their household environments, and behavioral factors relevant to health-seeking behaviors, allowing for a robust examination of determinants of delayed abortion seeking. Detailed distributions of these variables are presented in Table A in S1 Text and further described in the Results section.

## Target variable and features

The target variable in this study was the delay in seeking abortion, categorized as delayed (encoded 1 when abortions occurred after 12 weeks of pregnancy) or not delayed (encoded as 0 when it occurred before/at 12 weeks of pregnancy). This variable was derived from information on the timing of abortion-seeking behavior among reproductive-age women in Ethiopia.

The study included 16 features, which encompassed both individual and community level factors. These features included the woman's age, geographic region of residence, and whether she lived in an urban or rural area. Other features related to socio-economic status, such as the woman's highest level of education and wealth index, were also considered. Reproductive factors, such as the number of children the woman has ever given birth to, were included as well. Behavioral factors, such as whether the woman smoked cigarettes, consumed alcohol, or had regular media exposure, were also incorporated.

These features were selected for their potential relevance to understanding delays in seeking abortion, based on factors that may influence reproductive health and behavior in Ethiopia. More detailed information on the full list of features is available in Table A in S1 Text.

## Data preprocessing

The data preprocessing for this study was conducted in R, employing several key techniques to prepare the dataset for machine learning analysis. The initial step involved addressing missing values in specific variables, including media exposure, husband's education, and age. For handling missing data, KNN imputation was applied, which uses the k-nearest neighbors algorithm to impute missing values based on the values of nearby data points. Feature engineering was performed to create the 'Media Exposure' feature, derived from the frequency of watching TV and listening to the radio. Lumping was carried out for variables such as occupational status (for both the husband and the woman), total number of household members, total number of under-five children, wealth index, and total number of children ever born.

Feature selection was conducted using the Boruta algorithm, chosen for its robustness in identifying all relevant predictors by comparing the importance of original features with permuted shadow features. Boruta is particularly well-suited for health-related behavioral datasets because it reduces the risk of excluding weak but meaningful predictors, a limitation common in traditional filter-based methods. In the context of reproductive health research, previous studies have demonstrated that multivariable relationships such as women's autonomy, media exposure, fertility history, and socioeconomic status often interact in complex ways, therefore, a comprehensive wrapper-based approach such as Boruta is preferable for capturing non-linear and interaction-based effects reported in prior studies on delays in seeking abortion or maternal health services (Fig A in S1 Text).

We cross-referenced the selected predictors with evidence from the existing literature. Factors such as media exposure, parity, wealth index, women's education, and decision-making autonomy have consistently been reported as significant determinants of reproductive health service utilization and delays in care-seeking. The removal of features such as the household size and the husband's occupational status aligned with findings indicating that these variables are not consistently strong predictors of delays in abortion-related care when controlling for socioeconomic and autonomy-related factors [7,8].

As part of the validation process, the stability of the Boruta-selected features was assessed through repeated execution of the feature-selection procedure across 30 randomized subsamples of the data. A predictor was retained only if it was confirmed as "important" in at least 80% of the iterations, ensuring robustness against sampling variability. In addition, the selected features were further evaluated through model-based importance rankings generated from the trained classifiers, confirming that the Boruta-retained predictors consistently contributed to performance across models (Fig B in S1 Text).

Initially, the dataset contained 16 features. After applying Boruta and validating the selected predictors, 10 features were retained as the most relevant for predicting delayed abortion. The rejected features included the decision maker for not using contraception, husband's educational status, occupational status (for both the husband and the woman), total number of household members, and total number of under-five children.

One-hot encoding was then applied to the remaining categorical variables. The dummyVars function was used to perform the encoding. The entire dataset, excluding the target variable, was passed through this function to create binary columns for each category. This resulted in a dataset where categorical variables were represented as separate binary columns. The target variable was separated from the features to ensure it remained intact for modeling. After one-hot encoding, the dataset was restructured, with the features and target variable organized separately.

To further enhance the quality of the dataset, correlation analysis was performed on the feature set. Using the cor function, a correlation matrix was generated to identify highly correlated features. A correlation threshold of 0.8 was set, and the findCorrelation function was used to detect features with correlations above this threshold. These highly correlated features were removed from the dataset to avoid multicollinearity, which could negatively impact the performance

of certain machine learning algorithms. After one-hot encoding and correlation filtering, the dataset contained 38 encoded features.

The data was split into training and testing sets, with 70% of the data used for training and 30% reserved for testing. This split was performed to allow for model training on one subset of the data while providing an unbiased evaluation of the model's performance on unseen data.

Throughout the preprocessing, various functions from the caret and Boruta packages were utilized, including dummyVars, predict, and findCorrelation, to streamline the process and ensure the dataset was appropriately prepared for machine learning analysis. The model framework of the study is illustrated in Fig 1.

## Machine learning algorithms and performance evaluation

In this study, eight machine learning algorithms were employed to predict delays in seeking abortion, chosen for their ability to handle various data characteristics and to ensure robustness. These models included K-Nearest Neighbors (KNN), Logistic Regression (LR), Gradient Boosting Machine (GBM), Naive Bayes (NB), Random Forest (RF), Support Vector Machine (SVM), Extreme Gradient Boosting (XGB), and Light Gradient Boosting Machine (LGBM). These algorithms were selected because they cover linear, probabilistic, distance-based, margin-based, and ensemble tree-based approaches, allowing the study to compare performance across fundamentally different learning mechanisms. This diversity reduces reliance on any single modeling assumption and helps identify the algorithm best suited to the structure and complexity of the dataset.

K-Nearest Neighbors (KNN) and Logistic Regression (LR) were included to represent traditional baseline classifiers. KNN, implemented using the caret package with cross-validated selection of neighbors, classifies observations based on similarity in feature space. LR provides a transparent, interpretable probabilistic model and was implemented using glm with regularization to prevent overfitting. These models offer strong interpretability and serve as essential references against which more complex algorithms can be evaluated [9].

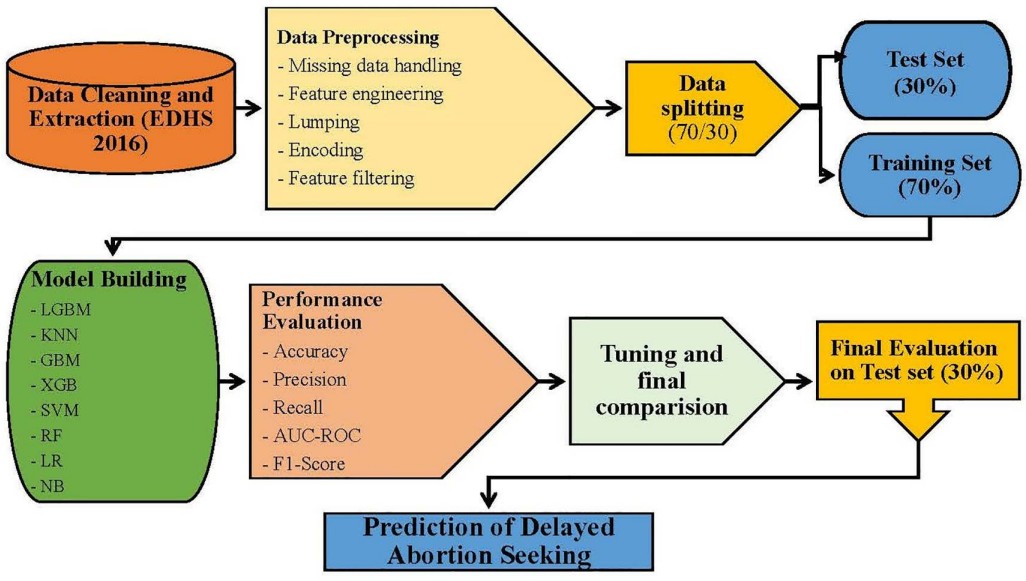

**Fig 1. Model Architecture for Predicting Delays in Seeking Abortion Among Reproductive-Aged Women in Ethiopia.**

Ensemble tree-based methods, Gradient Boosting Machine (GBM) and Random Forest (RF), were incorporated due to their proven ability to handle nonlinear relationships, interactions, and mixed-type features. GBM was trained using the gbm package with hyperparameter tuning via cross-validation, while RF constructed multiple randomized decision trees to improve predictive stability. These ensemble approaches are widely used in health data prediction because they manage complex structures with minimal preprocessing [10].

Naive Bayes (NB) and Support Vector Machine (SVM) models were were included to represent probabilistic and margin-based classifiers. NB, implemented using e1071, assumes conditional independence among predictors and performs well with high-dimensional or weakly correlated features. SVM, implemented via caret, used an RBF kernel to capture nonlinear boundaries and was optimized through hyperparameter tuning. These algorithms are valuable when relationships are not easily captured by linear models [11].

Finally, Extreme Gradient Boosting (XGB) and Light Gradient Boosting Machine (LGBM) were selected as advanced boosting algorithms known for their efficiency and strong performance on tabular datasets. XGB (via xgboost) and LGBM (via lightgbm) use gradient-boosted tree ensembles but differ in optimization strategy, enabling fast training and high accuracy. Their inclusion provided additional model diversity and allowed evaluation of state-of-the-art approaches commonly used in applied predictive modeling [12].

Model performance was assessed using accuracy, precision, recall, F1-score, and AUC to evaluate both discrimination and generalization. This multi-algorithm framework ensured a thorough comparison and supported the selection of the most reliable predictive model for delayed abortion seeking in Ethiopia.

## Hyperparameter tuning and model building

For the XGBoost (XGB) model, hyperparameters such as tree depth, minimum child weight, and learning rate were fine-tuned. The max_depth was varied from 6 to 14, and min_child_weight was tested between 0.1 and 1.0. The learning rate (eta) was explored from 0.01 to 0.2, and the gamma, which controls regularization, was adjusted between 0 and 1. Additionally, colsample_bytree, determining the fraction of features used for each tree, was varied from 0.5 to 1. The number of boosting iterations (nrounds) was increased to 6000, with early stopping applied after 20 rounds of no improvement. These parameters were optimized using the xgboost package, and model performance was evaluated based on the AUC of the ROC curve.

Similarly, for the Random Forest (RF) model, key hyperparameters such as the number of trees (ntrees), tree depth (max_depth), and sampling rate (sample_rate) were adjusted. The number of trees was tested between 50 and 100, while the tree depth ranged from 10 to 20. The sample_rate, controlling the proportion of data used for each tree, was varied from 0.7 to 0.9. Cross-validation with 10 folds was employed, with early stopping implemented after 50 rounds based on the RMSE metric. Hyperparameter tuning was executed using the h2o.randomForest function. In the case of Support Vector Machines (SVM), hyperparameters such as the cost parameter (C) and the kernel's gamma parameter were optimized. The C values were adjusted to balance the bias-variance trade-off, while gamma was varied to control the decision boundary's curvature. A 10-fold cross-validation was performed using the trainControl function, and the optimal values were selected based on the Accuracy and AUC of the ROC curve using the svmRadial method from the caret package.

Further, for the Naive Bayes (NB) model, cross-validation with 10 folds and 5 repetitions was utilized to determine optimal performance. The model's hyperparameters, particularly related to the distribution and smoothing, were adjusted using the train() function from the caret package, with performance evaluated based on accuracy. In the K-Nearest Neighbors (KNN) model, the number of neighbors (k) was explored through a grid search. Values of k ranged from 1 to a third of the dataset size. A 10-fold repeated cross-validation was used, and the optimal k was selected based on the AUC of the ROC curve. Hyperparameter tuning was performed using the train() function, with performance assessed through the twoClassSummary function. The Logistic Regression (LR) model involved repeated 10-fold cross-validation,

with hyperparameters focused on binomial classification. The model was trained using the glm function, selecting the best model based on accuracy and AUC metrics.

Lastly, for LightGBM (LGB), key hyperparameters like the learning rate, number of leaves, and tree depth were optimized. The learning rate was tested between 0.01 and 0.2, and the number of leaves varied from 20 to 40. The model's tree depth was allowed to grow freely with no limit, and early stopping was implemented after 10 rounds of no improvement in performance. The model was optimized for binary classification. Finally, for Gradient Boosting Machine (GBM), a grid search was conducted over learning rates ranging from 0.005 to 0.3. The number of trees and interaction depth were also fine-tuned. A total of 1000 trees were considered, and the optimal learning rate was chosen based on cross-validation results. Hyperparameter tuning was executed using the gbm function, with performance evaluated through RMSE and training time.

### Model interpretability

SHAP (Shapley Additive Explanations) values were used exclusively to interpret the contribution of each feature to the model predictions and were not employed to evaluate model performance, which was assessed using metrics such as accuracy, AUC, F1-score, sensitivity, and specificity. SHAP values provide a method for explaining individual predictions by attributing each feature's contribution to the final model output. This approach was particularly important in healthcare applications, where understanding model behavior is essential for decision-making. To visualize the contribution of each feature, two SHAP visualizations were employed: beeswarm and bar plots. These plots helped to highlight the influence of various features on the model's predictions [13].

The beeswarm plot was used to display how each feature's value impacted predictions across the dataset, with each point representing a feature's contribution. This allowed for the identification of key features that had a significant influence on the best model's predictions. The bar plot was used to summarize the average magnitude of these SHAP values, providing a clear ranking of feature importance. These visualizations helped in achieving transparency in the models, making it possible for healthcare professionals to better interpret the factors driving predictions for delayed abortion.

### Patient and public involvement

None.

## Result

### Sociodemographic characteristics

The prevalence of delayed abortion seeking was 1109 (54.3%). Women aged 35–39 represented the largest group in both categories, with 254 (27.2%) in the Not-Delayed group and 275 (24.8%) in the Delayed group. A higher proportion of women from rural areas experienced delays, accounting for 960 (86.6%) of the delayed cases. Educational level showed that 819 (73.9%) of delayed cases occurred among women with no formal education. Contraceptive decisions made jointly with a partner were common, especially in delayed cases, totaling 412 (56.6%). Additionally, husbands or partners with no education were more prevalent in delayed cases, comprising 535 (50.2%). Alcohol use was higher among women who delayed seeking abortion services, with 351 (31.7%) reporting alcohol consumption. Similarly, lack of media exposure was associated with delays, affecting 726 (65.5%) of the delayed group. Finally, delays were more frequent among women from poorer households, with 613 (55.3%) belonging to the delayed group. In the Somali region, 207 (18.7%) of women delayed seeking abortions, while Addis Ababa, an urban setting, saw fewer delays, with 27 (2.4%) cases (Table A in S1 Text).

### Hyperparameter tuning and training performance

The hyperparameter tuning process led to the identification of optimal configurations for each model, significantly enhancing their performance. For the XGB model, the optimal values were determined for several hyperparameters, including

a max_depth of 14, min_child_weight of 0.8, and a learning rate (eta) of 0.1. These values contributed to an exceptional accuracy of 95.6% (94.4%, 96.6%), with precision (96.7), and recall of (93.8), demonstrating strong performance.

Similarly, for the RF model, hyperparameters were tuned to achieve optimal results. The optimal configuration included a max_depth of 20, min_rows of 1, and 50 trees with a sample_rate of 0.8. This resulted in an impressive accuracy of 97.3% (96.3%, 98.1%), with a precision of 96.7, recall of 98.4, and F1-score of 97.5, underscoring its effectiveness. In contrast, the SVM model, after tuning its hyperparameters through 10-fold cross-validation, achieved 94.4% (93.0%, 95.5%) accuracy, a precision of 93.2, and recall was 96.6, reflecting strong performance, although slightly lower than RF and XGB.

The NB model, tuned using 10-fold cross-validation, showed 59.0% (56.4%, 61.6%) accuracy. Despite a relatively high precision of 76.3, its recall of 35.7 led to an F1-score of 48.6, indicating suboptimal performance in balancing precision and recall. For the KNN model, tuning the number of neighbors (k) led to an accuracy of 84.3% (82.4%, 86.2%). Precision and recall were balanced at 85.6 and 85.5, respectively. The F1-score was 85.5, reflecting a solid performance, though it did not surpass models like RF and XGB. The LR model, after hyperparameter tuning using 10-fold cross-validation, achieved an accuracy of 66.6% (64.1%, 69.0%). Precision was 67.7 and recall was 73.4, leading to an F1-score of 70.4, showing moderate performance. For the LGB model, the optimal hyperparameters included a learning rate of 0.1 and 31 leaves. This configuration resulted in an accuracy of 88.3% (86.5%, 89.9%), precision of 90.4 and a recall of 83.7. F1-score was 89.3, indicating strong performance, although not as high as RF or XGB.

Lastly, the GBM model, tuned using grid search, achieved an accuracy of 65.1% (62.6%, 67.6%). The AUC-ROC was 79.8, and while it exhibited a high recall of 96.9, its precision was lower at 61.3, resulting in an F1-score of 75.0, reflecting moderate performance. In summary, the Random Forest and XGBoost models outperformed others, achieving high accuracy, AUC-ROC, precision, recall, F1-score, and Kappa values. The Support Vector Machine and LightGBM models also performed competitively, while Logistic Regression and Naive Bayes performed more modestly in comparison (Table 1). The optimal hyperparameters and their values are provided in Table B in S1 Text.

## Performance on the test set

Fig 2 presents the performance metrics of eight machine learning models on the test set, used to predict delay in seeking abortion among reproductive-age women in Ethiopia. The models were evaluated based on accuracy, sensitivity/recall, precision/PPV, F1-score, and AUC-ROC. RF achieved the highest overall accuracy of 91.8% (95% CI: 89.3, 93.8), demonstrating strong performance across all metrics with balanced sensitivity and precision, making it the most reliable model for both detection and classification of delayed abortion seekers. SVM and XGB also performed well, with accuracy rates of 87.9% (95% CI: 85.0, 90.4) and 89.7% (95% CI: 87.0, 92.0), respectively, and strong F1-scores. XGB, in particular, demonstrated a good balance between sensitivity and precision, highlighting its effectiveness in both correctly identifying delayed abortion seekers and minimizing false positives. While the dataset in this study was balanced, XGB's robust performance suggests that it would also perform well in scenarios with class imbalance, where it could maintain this balance and handle the trade-off between sensitivity and precision effectively.

**Table 1. Performance metrics of each machine learning model on the training set.**

| ML model | Accuracy (95% CI) | SS/Recall | Precision/PPV | AUC-ROC | F1-score |
|---|---|---|---|---|---|
| KNN | 84.3% (82.4, 86.2) | 85.5 | 85.6 | 94.4 | 85.5 |
| LR | 66.6% (64.1, 69.0) | 73.4 | 67.7 | 72.5 | 70.4 |
| GBM | 65.1% (62.6, 67.6) | 96.9 | 61.3 | 79.8 | 75.0 |
| NB | 59.0% (56.4, 61.6) | 35.7 | 76.3 | 68.3 | 48.6 |
| RF | 97.3% (96.3, 98.1) | 98.4 | 96.7 | 99.5 | 97.5 |
| SVM | 94.4% (93.0, 95.5) | 96.6 | 93.2 | 97.5 | 94.8 |
| XGB | 95.6% (94.4, 96.6) | 93.8 | 96.7 | 99.4 | 95.9 |
| LGBM | 88.3% (86.5, 89.9) | 83.7 | 90.4 | 95.9 | 89.3 |

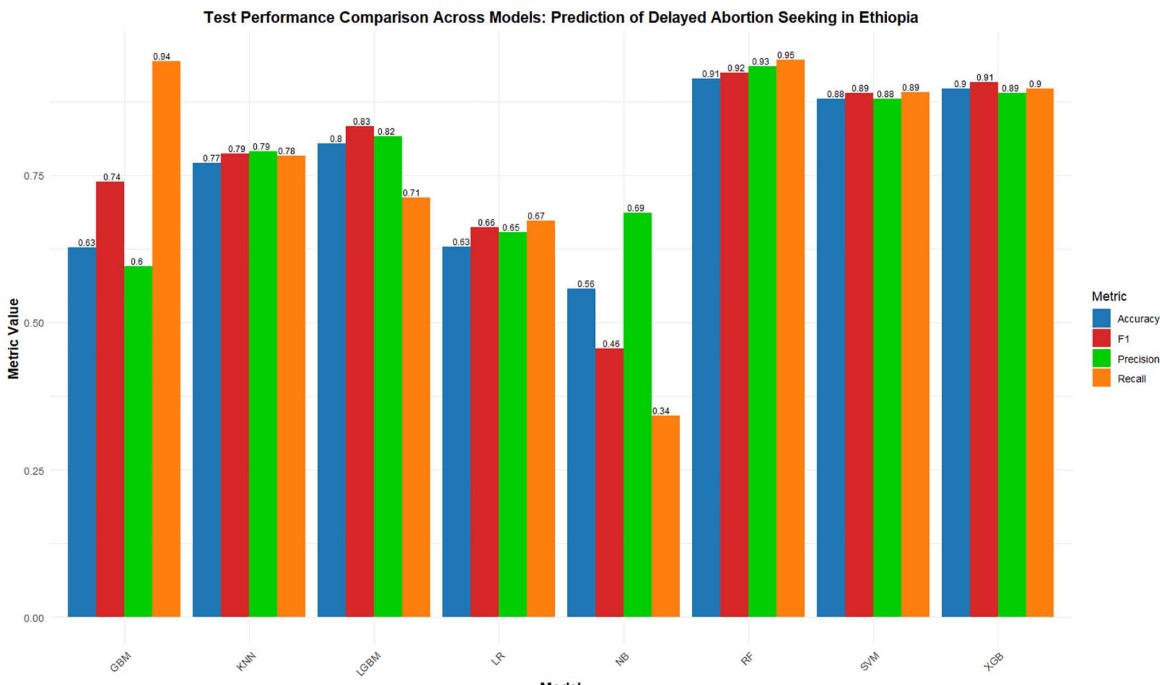

**Fig 2. Performance of machine learning models on the test set for predicting delay in seeking abortion among reproductive-age women in Ethiopia.**

In contrast, LR, GBM, NB, and KNN exhibited lower accuracy and mixed performance across the other metrics. Notably, while the NB model had relatively low accuracy (55.7%), it showed a higher precision score (68.6%) compared to its sensitivity, indicating that it correctly identified positive cases at a higher rate but struggled with recall. The LGB model achieved an accuracy of 80.4% (95% CI: 77.0, 83.4) with strong precision and F1-score values, although its recall was somewhat lower compared to the other high-performing models.

The Area Under the Receiver Operating Characteristic Curve (AUC-ROC) values for the eight machine learning models evaluated in this study is illustrated in Fig 3. The RF model achieved the highest AUC of 97.6, demonstrating excellent ability to distinguish between delayed and undelayed abortion seekers. Following closely were XGB and SVM, with AUC values of 96.6 and 94.7, respectively, indicating strong discriminatory power. The KNN model also showed a good AUC of 88.2, while the LGB model achieved a solid AUC of 89.8. On the other hand, LR, GBM, and NB had relatively lower AUC values of 67.6, 74.8, and 64.4, respectively, reflecting poorer ability to discriminate between the classes. These values provide further insight into the comparative performance of the models in terms of their classification abilities, particularly in distinguishing between true positives and false positives.

The learning curves for all proposed models illustrate how model accuracy varies with increasing fractions of the training data. These curves provide insight into the convergence behavior and generalization capacity of each algorithm, including RF, GBM, LightGBM, XGBoost, SVM, LR, KNN, and NB. By visualizing model performance across different training set sizes, the stability and robustness of the predictive models is highlighted. The learning curves are presented in a consolidated figure for ease of comparison and clarity (Fig C in S1 Text).

## Model interpretability

SHAP bar and BeeSwarm plots are presented for the Random Forest and XGBoost models, as these two models demonstrated the highest classification performance in terms of accuracy and AUC. Given their strong predictive capabilities,

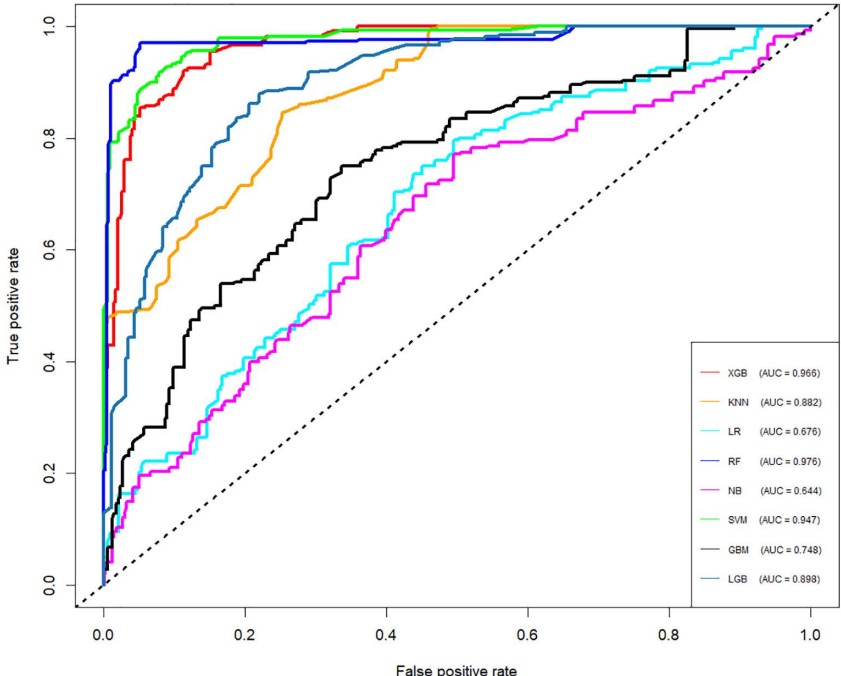

**Fig 3. Receiver Operating Characteristic (ROC) curves for the eight machine learning models for predicting delay in seeking abortion among reproductive-age women in Ethiopia.**

they are ideal candidates for interpreting the impact of individual features on model predictions. These plots provide valuable insights into feature importance and how each feature influences the models' outputs.

The SHAP plots for the RF model reveal the impact of various features on predicting the likelihood of delay in seeking an abortion among reproductive-aged women in Ethiopia, with positive and negative contributions indicated by the direction of the SHAP values. Features with higher SHAP values indicate stronger positive contributions to the prediction, while lower (negative) SHAP values suggest a negative impact on the likelihood of seeking an abortion in a timely manner.

Among the most influential features, women aged between 40–44 have the most significant positive impact on the prediction, with a SHAP value of 0.3385. This indicates that women in this age group are more likely to delay seeking an abortion. The distribution of feature values for this group shows a clear positive trend, with higher values extending towards the positive SHAP values, suggesting that as women's age falls within this range, the likelihood of delay in seeking an abortion increases.

The next most influential factor is living in the Somali region, with a SHAP value of 0.2521. Women from this region also have a higher likelihood of delaying an abortion. The concentration of higher values is strongly extended towards the positive SHAP values, indicating a consistent trend in this region. In addition, not having been exposed to media is another factor with a SHAP value of 0.2441, indicating that women who have not been exposed to media in the relevant context are more likely to delay seeking an abortion. The SHAP values for this feature show some mixture, but higher values predominantly push the model's prediction towards the positive SHAP values.

Alcohol consumption also has a positive effect, with a SHAP value of 0.2202, suggesting that women who drink alcohol are more likely to experience delays. The distribution for this feature is somewhat mixed, but the higher values lean towards the positive SHAP side, indicating that alcohol consumption generally increases the likelihood of delay in seeking

an abortion. Similarly, living in the Amhara region shows a positive impact with a SHAP value of 0.2150, meaning women from this region are more likely to delay seeking an abortion. The distribution is mostly positive, with higher SHAP values extending towards the positive end, though there is some variability in the data.

Women aged between 45–49 also exhibit a moderate positive influence, with a SHAP value of 0.1806, indicating a tendency for delay in seeking an abortion. The distribution for this feature, while mostly positive, shows some overlap with lower values, suggesting that the effect is not as consistent as with younger age groups. Wealthier women, classified as "richer," also show a slight positive contribution with a SHAP value of 0.1675, suggesting that wealthier women tend to delay seeking an abortion. The higher values for this feature are generally on the positive side, though some mixture in the distribution suggests that wealthier women are not uniformly more likely to delay.

In contrast, living in urban areas has a negative influence on the likelihood of delay, with a SHAP value of 0.2203, suggesting that urban residents are less likely to delay seeking an abortion. The SHAP values for this feature show a perfect negative trend, with points clustering towards the negative SHAP values, indicating that urban residence reduces the probability of delay in seeking an abortion. Lastly, living in Addis Ababa and the Southern Nations, Nationalities, and People's Region (SNNPR) both show negative influences on the likelihood of delay in seeking an abortion, with SHAP values of 0.1634 and 0.1461, respectively. Women from these regions are less likely to experience delays. The feature values for both regions show a strong negative trend, with higher values extending towards the negative SHAP values, indicating that these regions contribute to a lower likelihood of delay.

Overall, the SHAP plots of the RF model highlight features that significantly impact the likelihood of delay in seeking an abortion, with age, regional differences, and media exposure playing crucial roles. The distributions of feature values reveal both consistent trends and variability in their influence on delayed abortion seeking. However, for certain features such as women in their early thirties, those with primary education, individuals from the poorest wealth categories, and women from regions like Oromia, the SHAP values show considerable variability. These mixed distributions make it difficult to interpret their impact clearly, and relying on them for conclusions may lead to misleading interpretations. Thus, careful consideration is needed when interpreting these features (Fig 4).

The SHAP bar and beeswarm plots for the XGB model reveal key features that positively impact the likelihood of delay in seeking an abortion. Features such as women aged 40–44, living in the Somali region, and women aged 45–49 show positive contributions, with their high feature values (golden points) extending towards the positive SHAP values. Similarly, factors like lack of exposure to reproductive health media and alcohol consumption also have positive impacts, with the feature values pushing towards higher SHAP values, suggesting a higher likelihood of delay in seeking an abortion (Fig 6).

In contrast, living in Addis Ababa and residing in urban areas show negative impacts on the likelihood of delay, with their high feature values extending towards the negative SHAP values. The XGB model provides clear and consistent patterns in these influences, avoiding the mixed points often seen in the RF model. This clarity in the XGB plot further strengthens the interpretability of the model's predictions. The overall trends are similar to those of the RF model, with both models identifying age and regional factors as key influencers, while urban residence and living in Addis Ababa are associated with a lower likelihood of delay (Fig 5).

## Discussion

This study aimed to predict delays in seeking abortion services among reproductive-aged women in Ethiopia using interpretable machine learning models, specifically focusing on the Random Forest (RF) and XGBoost (XGB) models. These models demonstrated superior performance, with both high accuracy and AUC scores, and were selected for SHAP (SHapley Additive exPlanations) analysis to interpret feature contributions. The analysis identified key factors influencing delayed abortion-seeking behavior, such as age, region of residence, media exposure, alcohol consumption, and urban versus rural residence. The findings offer valuable insights into the socio-demographic factors driving delayed

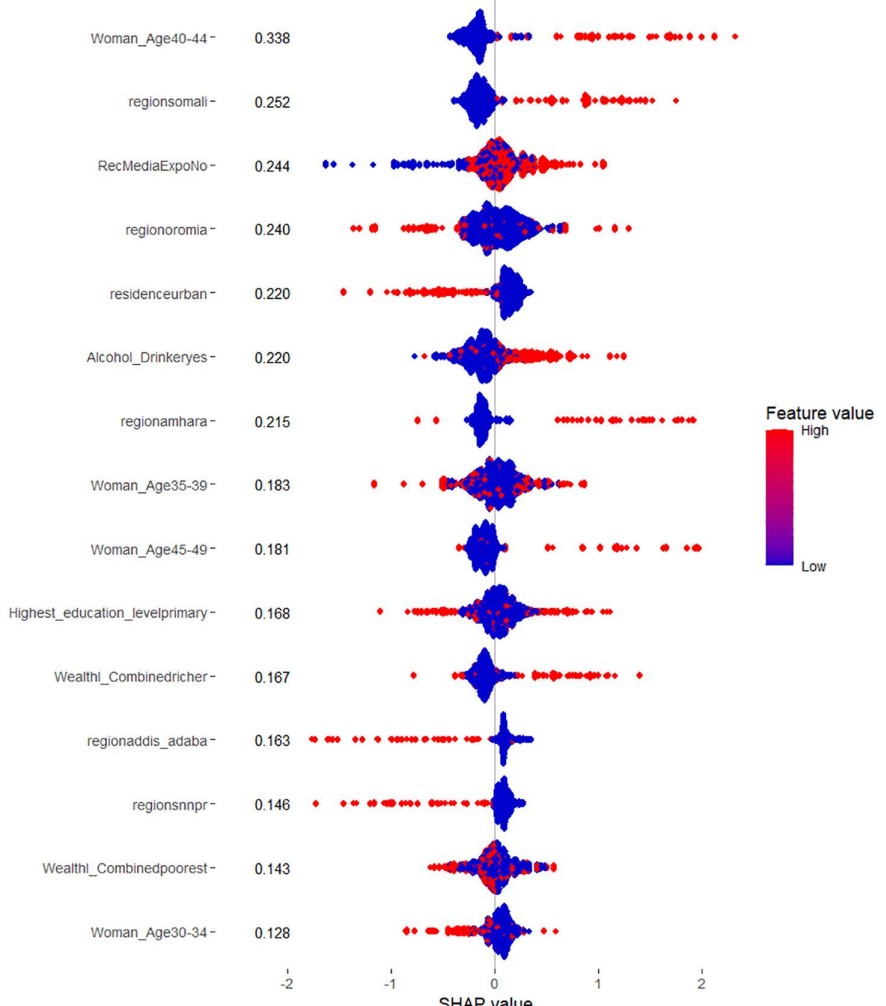

**Fig 4. SHAP BeeSwarm Plot of the RF model for prediction of delay in seeking abortion among reproductive-age women in Ethiopia.**

access to abortion services, and these results carry important implications for both public health interventions and policy development.

Age was one of the most influential features in both the RF and XGB models. Women aged 40–44 had the highest positive SHAP values, indicating that older women in this age group are significantly more likely to experience delays in seeking abortion services. Women aged 45–49 also exhibited a positive influence on delay, although to a lesser extent. This finding is consistent with previous studies, which suggest that older women may face greater stigma or social barriers when seeking abortion, leading to delays. Additionally, older women may be more hesitant to seek timely abortion services due to misconceptions about their health or reproductive rights. Importantly, younger women below the age of 40 were less likely to delay abortion, highlighting the need for targeted interventions aimed at older women, who may require additional support or counseling [14,15].

Regional differences were another important features influencing abortion-seeking delays. Women residing in the Somali region were shown to have a higher likelihood of delay in both models, with SHAP values strongly indicating this positive association. This may reflect the region's more conservative social norms, limited access to reproductive

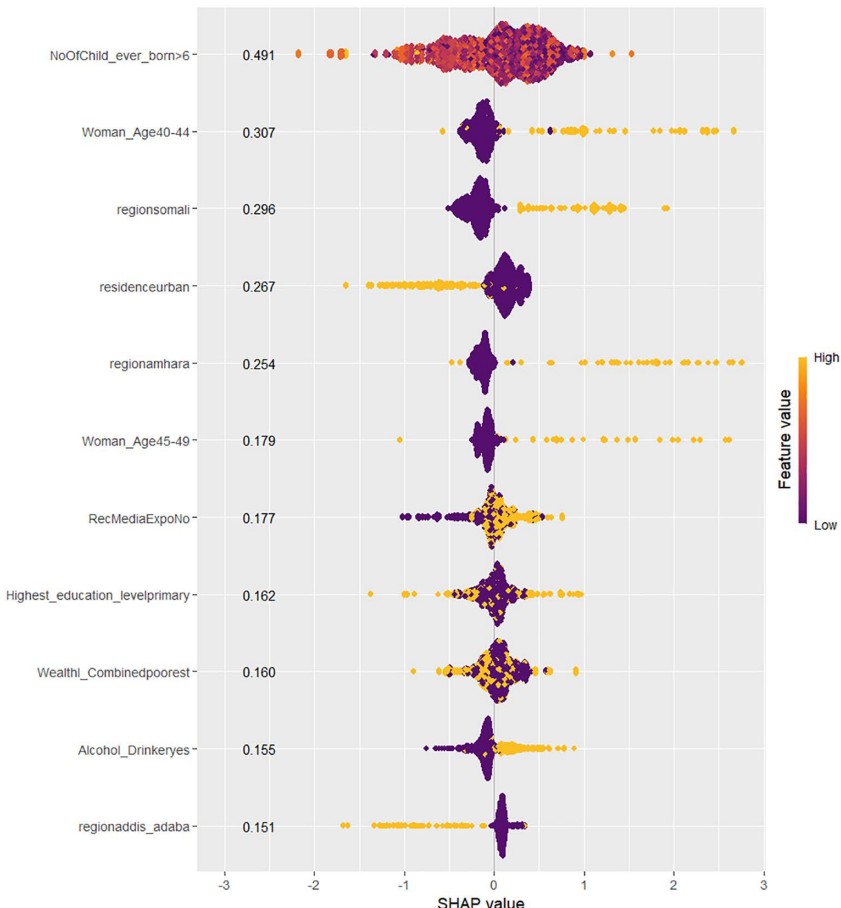

**Fig 5. SHAP BeeSwarm Plot of the XGB model for prediction of delay in seeking abortion among reproductive-age women in Ethiopia.**

health services, or cultural barriers that prevent timely access to abortion. Similarly, women from the Amhara region also exhibited a positive association with delay, suggesting that regional disparities in healthcare infrastructure and cultural attitudes towards abortion might contribute to delays. Conversely, women living in Addis Ababa and the Southern Nations, Nationalities, and People's Region (SNNPR) had negative SHAP values, indicating that they were less likely to delay seeking abortion services. These regions, particularly Addis Ababa, are known for having better access to healthcare facilities and more progressive attitudes toward reproductive health, which likely reduces delays. This finding underscores the importance of addressing regional inequalities in healthcare access to ensure that women in more rural or conservative areas, such as the Somali and Amhara regions, receive timely abortion services [3,16].

Lack of exposure to reproductive health media also had a significant positive impact on the likelihood of delay in both the RF and XGB models. Women who had not been exposed to media campaigns or educational programs related to reproductive health were more likely to experience delays in seeking abortion. This finding highlights the crucial role that media plays in disseminating information about reproductive health services. In Ethiopia, where media campaigns often serve as an important source of information on healthcare services, women who lack access to these campaigns may be unaware of their reproductive rights or the availability of safe abortion services. Enhancing media outreach, particularly in rural areas and regions with conservative social norms, could help reduce delays in seeking abortion services by improving women's knowledge and awareness [17,18].

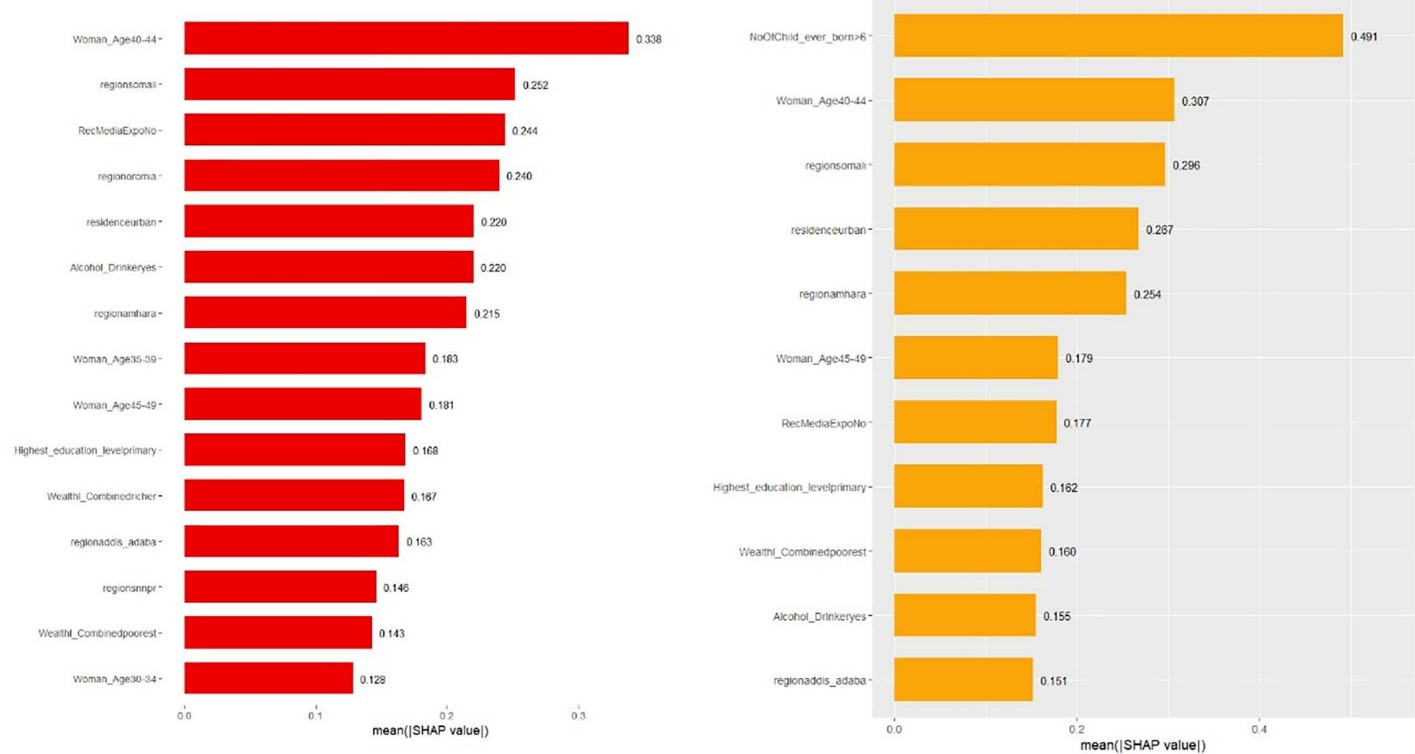

**Fig 6. SHAP Bar Plots of the RF and XGB models for prediction of delay in seeking abortion among reproductive-age women in Ethiopia.**

Alcohol consumption was another factor positively associated with delays in seeking abortion services. Women who consumed alcohol were more likely to delay, as indicated by the positive SHAP values for this feature. This may be due to a variety of socio-behavioral factors, including the potential for alcohol to impair decision-making, exacerbate feelings of stigma, or create additional barriers to healthcare access. Women who drink alcohol may also face greater social judgment or discrimination, leading to further delays in seeking medical care. Public health interventions could focus on providing targeted reproductive health services to women who consume alcohol, addressing both the behavioral and social challenges they face in accessing timely care [19,20].

Residence was another important factor influencing delays. Women living in urban areas, particularly in Addis Ababa, were less likely to experience delays, as indicated by the strong negative SHAP values in both the RF and XGB models. Urban women generally have better access to healthcare services, greater exposure to media campaigns, and more progressive social norms surrounding reproductive health, all of which contribute to more timely abortion-seeking behavior. In contrast, rural women, particularly those living in regions like Amhara and Somali, are more likely to face significant barriers to accessing abortion services, including limited healthcare infrastructure, cultural taboos, and lack of information. This urban-rural divide highlights the importance of improving healthcare access in rural areas and addressing the socio-cultural factors that contribute to delays in abortion-seeking behavior [21,22].

Beyond these quantitative predictors, qualitative insights from previous studies suggest that women's abortion-seeking timelines are shaped by complex emotional, social, and interpersonal factors that are not fully captured in structured survey variables. Many women experience uncertainty about the pregnancy, fear of social judgment, or reluctance to disclose the pregnancy to partners or family members, which can lead to prolonged decision-making. In settings where

abortion carries moral stigma or is perceived as socially unacceptable, women may delay seeking care while attempting to reconcile personal beliefs with external expectations. Additionally, limited autonomy in reproductive decision-making, particularly among married or rural women, may require negotiating with partners or elders before accessing services, which further contributes to delays. These qualitative dimensions complement the model's findings and highlight that delays often arise not only from demographic factors, but also from broader social norms, gender dynamics, and personal coping processes that influence how and when women seek abortion care [2,17,23,24].

This study makes novel contributions to the existing literature on abortion-seeking delays in low-resource settings. First, to our knowledge, this is the first study to employ interpretable machine-learning models, specifically Random Forest and XGBoost combined with SHAP explainability, to analyze determinants of delayed abortion seeking among Ethiopian women using a large, nationally representative dataset. While previous studies have relied primarily on traditional regression and pure qualitative approaches, the use of interpretable ML allows for capturing complex, nonlinear relationships and variable interactions that conventional models often overlook. Second, the study advances methodological practice by systematically comparing multiple machine-learning models to identify the most accurate and reliable predictors of delay, thereby strengthening the robustness of the findings. Third, the integration of model interpretability through SHAP values offers a transparent and user-friendly explanation of each predictor's directional influence, bridging the gap between advanced algorithms and practical decision-making. Collectively, these methodological innovations contribute new analytical tools to reproductive health research in Ethiopia and provide a replicable framework for future studies in similar contexts.

In summary, the findings highlight the critical role that sociodemographic factors, such as age, region, and media exposure, play in influencing delays in seeking abortion services. These factors were consistently identified as key contributors across different machine learning models, particularly in the context of Ethiopian women. Addressing these influences through targeted interventions could significantly improve timely access to safe abortion services and reduce the associated risks of delayed procedures.

## Strength and limitations

This study has several notable strengths, including the analysis based on a comprehensive, nationally representative data, the Ethiopian Demographic and Health Survey (EDHS) 2016, ensuring that the findings are reflective of the broader population of reproductive-aged women in Ethiopia. The use of advanced machine learning models, such as Random Forest and XGBoost, along with SHAP analysis, provided both robust predictions and interpretable insights into feature importance, enhancing the study's explanatory power.

Despite these strengths, there are limitations to consider. The reliance on self-reported data for factors such as media exposure, alcohol consumption, and the timing of abortion-seeking introduces potential biases. These biases may arise from memory recall issues, social desirability bias, or inaccuracies in timing, potentially affecting the reliability of the results. Additional qualitative research could also help explore the underlying reasons why factors such as alcohol consumption and regional disparities have such a strong influence on delays in abortion-seeking behavior.

Future research could also expand on this study by investigating the impact of specific interventions, such as media campaigns or regional healthcare improvements, on reducing delays. Moreover, exploring the role of other socio-economic and psychological factors in influencing delays could provide a more comprehensive understanding of the barriers women face in accessing abortion services.

## Conclusion and recommendation

This study highlights several key factors influencing delays in seeking abortion services in Ethiopia, including age, regional disparities, media exposure, and alcohol consumption. Among the machine learning models used, the Random Forest (RF) model was the best performer in predicting delays in seeking abortion services. The SHAP analysis provided clear

insights into how these features contribute to delays, offering valuable information for designing targeted interventions to improve access to reproductive health services. By addressing the socio-cultural and regional barriers identified in this study, public health efforts can work towards reducing maternal morbidity and mortality associated with unsafe abortions.

Building on these insights, the findings of this study have several important implications for public health initiatives and policy development. First, targeted interventions should be developed to support older women, particularly those in the 40–49 age group, who are at higher risk of delaying abortion. This could include counseling services, community outreach programs, and educational campaigns aimed at reducing stigma and increasing awareness of the importance of timely access to abortion services.

Second, addressing regional disparities is critical. In regions like Somali and Amhara, where delays are more common, improving healthcare infrastructure, increasing the availability of reproductive health services, and conducting culturally sensitive education campaigns could help mitigate delays. Policymakers should also focus on expanding access to safe abortion services in these regions by reducing legal, financial, and social barriers. Third, expanding media outreach and educational campaigns is essential to reduce delays caused by a lack of knowledge about reproductive health services. Media campaigns should be tailored to reach women in rural and conservative areas, providing them with clear information on their reproductive rights and the availability of safe abortion services.

Finally, public health interventions should be designed to support women who consume alcohol, addressing the unique barriers they face in accessing healthcare. These programs could include targeted counseling and support services that tackle both social stigma and behavioral factors associated with alcohol use and healthcare access. Moreover, while our study provided important quantitative evidence, future qualitative research is essential to capture the contextual and behavioral factors influencing women's decisions, dimensions that large-scale survey data alone cannot fully reflect.

## Supporting information

**S1 Text. Supplementary files for Interpretable Machine Learning for Predicting Delays in Seeking Abortion Among Reproductive-Aged Women in Ethiopia: A Study Using EDHS 2016 Data.**
(DOCX)

## Author contributions

**Conceptualization:** Meron Asmamaw Alemayehu, Almaw Genet Yeshiwas, Chalachew Yenew, Berhanu Abebaw Mekonnen.

**Data curation:** Meron Asmamaw Alemayehu, Almaw Genet Yeshiwas, Abebaw Molla Kebede, Habitamu Mekonen, Getaneh Atikilt Yemata, Amare Genetu Ejigu, Ahmed Fentaw Ahmed, Abathun Temesgen, Gashaw Melkie Bayeh, Chalachew Yenew, Rahel Mulatie Anteneh, Zeamanuel Anteneh Yigzaw, Fantu Mamo Aragaw, Getasew Yirdaw, Sintayehu Simie Tsega, Anley Shiferaw Enawgaw, Tilahun Degu Tsega, Berhanu Abebaw Mekonnen.

**Formal analysis:** Meron Asmamaw Alemayehu, Getaneh Atikilt Yemata, Amare Genetu Ejigu, Abathun Temesgen, Chalachew Yenew, Zeamanuel Anteneh Yigzaw, Fantu Mamo Aragaw, Anley Shiferaw Enawgaw, Zufan Alamrie Asmare, Berhanu Abebaw Mekonnen.

**Funding acquisition:** Meron Asmamaw Alemayehu, Amare Genetu Ejigu.

**Investigation:** Meron Asmamaw Alemayehu, Almaw Genet Yeshiwas, Abebaw Molla Kebede, Habitamu Mekonen, Getaneh Atikilt Yemata, Ahmed Fentaw Ahmed, Abathun Temesgen, Gashaw Melkie Bayeh, Chalachew Yenew, Rahel Mulatie Anteneh, Zeamanuel Anteneh Yigzaw, Fantu Mamo Aragaw, Getasew Yirdaw, Sintayehu Simie Tsega, Tilahun Degu Tsega, Zufan Alamrie Asmare, Berhanu Abebaw Mekonnen.

**Methodology:** Meron Asmamaw Alemayehu, Almaw Genet Yeshiwas, Abebaw Molla Kebede, Habitamu Mekonen, Getaneh Atikilt Yemata, Amare Genetu Ejigu, Gashaw Melkie Bayeh, Rahel Mulatie Anteneh, Fantu Mamo Aragaw, Getasew Yirdaw, Sintayehu Simie Tsega, Anley Shiferaw Enawgaw, Zufan Alamrie Asmare, Berhanu Abebaw Mekonnen.

**Project administration:** Meron Asmamaw Alemayehu, Almaw Genet Yeshiwas, Abebaw Molla Kebede, Habitamu Mekonen, Getaneh Atikilt Yemata, Amare Genetu Ejigu, Ahmed Fentaw Ahmed, Abathun Temesgen, Gashaw Melkie Bayeh, Chalachew Yenew, Rahel Mulatie Anteneh, Zeamanuel Anteneh Yigzaw, Getasew Yirdaw, Sintayehu Simie Tsega, Anley Shiferaw Enawgaw, Tilahun Degu Tsega, Zufan Alamrie Asmare.

**Resources:** Meron Asmamaw Alemayehu, Abebaw Molla Kebede, Habitamu Mekonen, Amare Genetu Ejigu, Ahmed Fentaw Ahmed, Abathun Temesgen, Gashaw Melkie Bayeh, Rahel Mulatie Anteneh, Zeamanuel Anteneh Yigzaw, Fantu Mamo Aragaw, Getasew Yirdaw, Sintayehu Simie Tsega, Anley Shiferaw Enawgaw, Tilahun Degu Tsega, Zufan Alamrie Asmare, Berhanu Abebaw Mekonnen.

**Software:** Meron Asmamaw Alemayehu, Ahmed Fentaw Ahmed, Abathun Temesgen, Chalachew Yenew, Zeamanuel Anteneh Yigzaw, Fantu Mamo Aragaw, Sintayehu Simie Tsega, Anley Shiferaw Enawgaw, Tilahun Degu Tsega, Berhanu Abebaw Mekonnen.

**Supervision:** Meron Asmamaw Alemayehu, Habitamu Mekonen, Ahmed Fentaw Ahmed, Rahel Mulatie Anteneh, Zeamanuel Anteneh Yigzaw, Fantu Mamo Aragaw, Getasew Yirdaw, Sintayehu Simie Tsega, Anley Shiferaw Enawgaw, Tilahun Degu Tsega, Zufan Alamrie Asmare, Berhanu Abebaw Mekonnen.

**Validation:** Meron Asmamaw Alemayehu, Almaw Genet Yeshiwas, Ahmed Fentaw Ahmed, Gashaw Melkie Bayeh, Fantu Mamo Aragaw, Getasew Yirdaw, Sintayehu Simie Tsega, Anley Shiferaw Enawgaw, Tilahun Degu Tsega, Berhanu Abebaw Mekonnen.

**Visualization:** Meron Asmamaw Alemayehu, Almaw Genet Yeshiwas, Abebaw Molla Kebede, Getaneh Atikilt Yemata, Gashaw Melkie Bayeh, Getasew Yirdaw, Anley Shiferaw Enawgaw.

**Writing – original draft:** Meron Asmamaw Alemayehu.

**Writing – review & editing:** Meron Asmamaw Alemayehu, Almaw Genet Yeshiwas, Abebaw Molla Kebede, Habitamu Mekonen, Getaneh Atikilt Yemata, Amare Genetu Ejigu, Ahmed Fentaw Ahmed, Abathun Temesgen, Gashaw Melkie Bayeh, Chalachew Yenew, Rahel Mulatie Anteneh, Zeamanuel Anteneh Yigzaw, Fantu Mamo Aragaw, Getasew Yirdaw, Sintayehu Simie Tsega, Anley Shiferaw Enawgaw, Tilahun Degu Tsega, Zufan Alamrie Asmare, Berhanu Abebaw Mekonnen.

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
