## [Decision Letter · Decision Letter 0]

10 Jul 2025

Response to Reviewers
Revised Manuscript with Track Changes
Manuscript
**Journal Requirements:**
**Additional Editor Comments (if provided):**
**Reviewers' Comments:**

**Comments to the Author**

1. Does this manuscript meet PLOS Digital Health’s publication criteria?

Reviewer #1: Partly

Reviewer #2: Yes

2. Has the statistical analysis been performed appropriately and rigorously?

Reviewer #1: Yes

Reviewer #2: Yes

3. Have the authors made all data underlying the findings in their manuscript fully available (please refer to the Data Availability Statement at the start of the manuscript PDF file)?

Reviewer #1: Yes

Reviewer #2: Yes

4. Is the manuscript presented in an intelligible fashion and written in standard English?

Reviewer #1: No

Reviewer #2: Yes

Reviewer #1: Thank you to look over my insight full expertise on this research work entitled “Interpretable Machine Learning for Predicting Delays in Seeking Abortion Among Reproductive-Aged Women in Ethiopia: : A Study Using EDHS 2016 Data” and I got this article is very interesting and thought full and identifying deferent Hedin factors that why mothers delay in Seeking Abortion service.

The research work is very interesting and thought full for scholars and also leading factor for other researchers

Saying this, I have seen the title “Delays in seeking abortion and its determinants among reproductive-age women based on the Ethiopian Demographic and health survey”

I do have some clarification about this manuscript and your research work?

1. In line 101: Various barriers continue to prevent women from seeking timely abortion care. These barriers include “restrictive legal frameworks”, how the restrictive legal framework is was barrier for delay for seeking abortion

2. Why you used 8 machine learning models and which problem is solved by using those models as machine learning researcher why not taking 4 or 5 models enough?

3. You and your team checks” Shapley Additive Explanations” in your research work, so which pattern or feature is more important and how do you measure.

4. Did you use SHAP for measure the performance of deferent models if yes how and no why?

Generally I have got the research work is important and thought full. But the document is better to upload in word document to address comments easily.

Thank you

Reviewer #2: This study identifies key factors influencing delays in seeking abortion services in Ethiopia. The authors of this work claimed that the study offers valuable insights into the socio-demographic factors driving delayed access to abortion services by women aged 45-49 and these results carry important implications for both public health interventions and policy development in Ethiopia. Particularly, the findings offer valuable insights for designing public health initiatives aimed at reducing unsafe abortion-related maternal morbidity and mortality. However, I suggest the authors to address following concerns in order to enhance the quality of this manuscript.

1. How feature selections were carried out? In this issue, is there any support from previous work or how the authors have validated the feature selection criteria? Please mention the process evidence-based it way.

2. Why eight machine learning algorithms were selected? Please justify.

3. Please describe more about data set and its characteristics in the manuscript.

4. Please present some qualitative results as the behavior of a pregnant woman couldn't be determined by quantitative aspects only.

5.How this work is novel in terms of data curation, proposed methodology, optimizing existing algorithm and so on?

6. Please interpret your result with explainable AI and also show the learning curve of the method.

In summary, the topic of this study and the issue are relevant. But, the contribution of the authors are limited to:

a. using existing dataset,

b. applying machine learning algorithms on them, and

c. suggested some recommendations.

Please show the novelty of this research. This study is also missing with a new framework to handle the identified issues.

**Do you want your identity to be public for this peer review?** For information about this choice, including consent withdrawal, please see our Privacy Policy

Reviewer #1: No

Reviewer #2: **Yes:** Rabindra Bista

**Figure resubmission:****Reproducibility:** To enhance the reproducibility of your results, we recommend that authors of applicable studies deposit laboratory protocols in protocols.io, where a protocol can be assigned its own identifier (DOI) such that it can be cited independently in the future. Additionally, PLOS ONE offers an option to publish peer-reviewed clinical study protocols. Read more information on sharing protocols at https://plos.org/protocols?utm_medium=editorial-email&utm_source=authorletters&utm_campaign=protocols

---

## [Editor Report · Decision Letter 1]

6 Nov 2025

Response to Reviewers
Revised Manuscript with Track Changes
Manuscript
**Journal Requirements:**
**Additional Editor Comments (if provided):**

Feature Selection: Explain how features were selected, supported by evidence or prior studies, and describe the validation process used.

Algorithm Choice: Justify the selection of the eight machine learning algorithms applied in the study.

Dataset Description: Provide a more detailed description of the dataset, including its characteristics and relevant attributes.

Qualitative Insights: Include qualitative analysis or discussion to complement the quantitative findings, especially considering that behavioral aspects (e.g., of pregnant women) may not be fully captured numerically.

Novelty and Contribution: Clearly articulate the novelty of the work—whether in data curation, methodological innovation, or algorithmic optimization.

Explainability and Learning Curves: Interpret results using explainable AI techniques and present the learning curves for the proposed models.

**Reviewers' Comments:**
**Figure resubmission:**

**Reproducibility:** To enhance the reproducibility of your results, we recommend that authors of applicable studies deposit laboratory protocols in protocols.io, where a protocol can be assigned its own identifier (DOI) such that it can be cited independently in the future. Additionally, PLOS ONE offers an option to publish peer-reviewed clinical study protocols. Read more information on sharing protocols at https://plos.org/protocols?utm_medium=editorial-email&utm_source=authorletters&utm_campaign=protocols

---

## [Decision Letter · Decision Letter 2]

20 Feb 2026

Interpretable Machine Learning for Predicting Delays in Seeking Abortion Among Reproductive-Aged Women in Ethiopia: A Study Using EDHS 2016 Data

PDIG-D-25-00032R2

Dear Ms. Alemayehu,

We are pleased to inform you that your manuscript 'Interpretable Machine Learning for Predicting Delays in Seeking Abortion Among Reproductive-Aged Women in Ethiopia: A Study Using EDHS 2016 Data' has been provisionally accepted for publication in PLOS Digital Health.

Best regards,

John Batani, Ph.D

Academic Editor

PLOS Digital Health

**Additional Editor Comments (if provided):**

The reviewers are happy with the revised version

**Reviewer Comments (if any, and for reference):**

Reviewer's Responses to Questions

**Comments to the Author**

Reviewer #1: All comments have been addressed

Reviewer #3: All comments have been addressed

Reviewer #4: (No Response)

publication criteria?

Reviewer #1: Yes

Reviewer #3: Yes

Reviewer #4: (No Response)

3. Has the statistical analysis been performed appropriately and rigorously?

Reviewer #1: I don't know

Reviewer #3: Yes

Reviewer #4: (No Response)

4. Have the authors made all data underlying the findings in their manuscript fully available (please refer to the Data Availability Statement at the start of the manuscript PDF file)?

Reviewer #1: Yes

Reviewer #3: Yes

Reviewer #4: No

5. Is the manuscript presented in an intelligible fashion and written in standard English?

Reviewer #1: Yes

Reviewer #3: (No Response)

Reviewer #4: No

Reviewer #1: i have got this research work gave sound for abortion delay and this finding might be used for policy makers

Reviewer #3: Accept

Reviewer #4: The revisions are acceptable.

**Do you want your identity to be public for this peer review?** For information about this choice, including consent withdrawal, please see our Privacy Policy

Reviewer #1: No

Reviewer #3: No

Reviewer #4: **Yes:** Mustafa Ghaderzadeh
